# Multication perovskite 2D/3D interfaces form via progressive dimensional reduction

Andrew H. Proppe [1,2,10], Andrew Johnston [2,10], Sam Teale[2,10], Arup Mahata [3,4], Rafael Quintero-Bermudez[2], Eui Hyuk Jung[2], Luke Grater[2], Teng Cui [5], Tobin Filleter [5], Chang-Yong Kim [6], Shana O. Kelley [1,7], Filippo De Angelis[3,4,8,9] & Edward H. Sargent [2✉]

Many of the best-performing perovskite photovoltaic devices make use of 2D/3D interfaces, which improve efficiency and stability – but it remains unclear how the conversion of 3D-to-2D perovskite occurs and how these interfaces are assembled. Here, we use in situ Grazing-Incidence Wide-Angle X-Ray Scattering to resolve 2D/3D interface formation during spin-coating. We observe progressive dimensional reduction from 3D to $n = 3 \rightarrow 2 \rightarrow 1$ when we expose $(MAPbBr_3)_{0.05}(FAPbI_3)_{0.95}$ perovskites to vinylbenzylammonium ligand cations. Density functional theory simulations suggest ligands incorporate sequentially into the 3D lattice, driven by phenyl ring stacking, progressively bisecting the 3D perovskite into lower-dimensional fragments to form stable interfaces. Slowing the 2D/3D transformation with higher concentrations of antisolvent yields thinner 2D layers formed conformally onto 3D grains, improving carrier extraction and device efficiency (20% 3D-only, 22% 2D/3D). Controlling this progressive dimensional reduction has potential to further improve the performance of 2D/3D perovskite photovoltaics.

---

[1] Department of Chemistry, University of Toronto, Toronto, ON, Canada. [2] The Edward S. Rogers Department of Electrical and Computer Engineering, University of Toronto, Toronto, ON, Canada. [3] D3-Computation, Istituto Italiano di Tecnologia, Genova, Italy. [4] Computational Laboratory for Hybrid/Organic Photovoltaics (CLHYO), Istituto CNR di Scienze e Tecnologie Chimiche (CNR-SCITEC), Istituto CNR di Scienze e Tecnologie Molecolari (ISTM-CNR), Perugia, Italy. [5] Department of Mechanical and Industrial Engineering, Toronto, ON, Canada. [6] Canadian Light Source, Saskatoon, SK, Canada. [7] Department of Pharmaceutical Sciences, Leslie Dan Faculty of Pharmacy, University of Toronto, Toronto, ON, Canada. [8] Department of Chemistry, Biology and Biotechnology, University of Perugia, Perugia, Italy. [9] Chemistry Department, College of Science, King Saud University, Riyadh, Saudi Arabia. [10]These authors contributed equally: Andrew H. Proppe, Andrew Johnston, Sam Teale. ✉email: ted.sargent@utoronto.ca

**2D**/3D metal halide perovskite solar cells make use of active layers comprised of 3D perovskites capped by a thin layer of 2D or quasi-2D perovskites—also known as reduced-dimensional perovskites (RDPs). Devices based on these materials have been shown to have superior stability imparted by the RDP capping layer[1–3]: RDPs exhibit improved stability relative to 3D analogs due to higher formation energies and favorable intermolecular bonding between ligand molecules[4], their ability to suppress ion migration[5,6], and hydrophobic ligands that can repel water[7]. 2D/3D interfaces can also improve fill factor and open-circuit voltage by passivating trap states, improving carrier extraction, and altering band alignment[8–10].

2D/3D interface formation is predominantly achieved through solution-processing (vacuum-processing has also been demonstrated[11]), in which a 3D perovskite layer is first deposited and annealed, and the surface is then exposed to a solution containing a low concentration of large ammonium cations (also referred to as ligand molecules) which transform part of the surface into RDPs (Fig. 1a). The thickness of these RDPs is designated by the number of layers (n) of perovskite octahedra that are enclosed by the ligand cations in the crystal structure. Purely 2D (n = 1) RDPs are almost ubiquitously formed in 2D/3D structures, with smaller proportions of n = 2 often observed as well[1,2,12]. The crystallinity and quality of the interface is also greatly affected by solvent: mixtures of isopropanol (needed to dissolve the ligand) and nonpolar solvents like chlorobenzene or chloroform are often used, although the isopropanol component can cause some redissolution of the 3D perovskite interface that is deleterious to performance[13].

Despite the widespread use of 2D/3D interfaces to augment perovskite device performance, their formation mechanism remains unclear. The perovskite lattice components are known to be highly labile[14,15], and cation exchange and insertion reactions can occur at room temperature[16], including full conversion of pure PbI₂ into n = 1 or n = 2 by exposure to ligand solutions[17], suggesting a similarly low activation barrier for 2D/3D interface formation—but how the 3D perovskite is fragmented into low-n RDPs, and whether this proceeds through intermediate states, is unknown.

We sought to gain insight into the formation mechanism of 2D/3D interfaces using in situ grazing-incidence wide-angle X-ray scattering (GIWAXS), which can inform on the orientation and d-spacing of any diffractive species that form during the spin-coating of the ligand solution. In situ GIWAXS has been used previously to study the formation of 2D or 3D perovskites directly from precursors in solution, tracking the formation of intermediate states and precursor complexes that eventually transform into perovskite[18–21]. Here we instead use this technique to probe the transformation of the already fully-formed 3D perovskite surface into RDPs, by collecting diffraction patterns before, during, and after exposure of the 3D perovskite to a solution containing 2D ligand cations.

## Results and discussion

**XRD and in situ GIWAXS.** We used the ligand cation 4-vinylbenzylammonium bromide (VBABr, Fig. 1b), which we have shown forms 2D/3D interfaces with RDPs that exhibit strong diffraction, and greatly improves the efficiency and stability of photovoltaic devices[10,12]. To obtain q values for the diffraction peaks of low-n RDPs, we first spin-coated thin films of ⟨n⟩ = 1, 2, and 3 directly from RDP precursor solutions using the composition $(VBABr)_2(FA_{0.95}MA_{0.05})_{n-1}Pb_n(I_{0.95}Br_{0.05})_{3n+1}$, which has the same cation and anion composition as the 2D/3D films used throughout this work. X-ray diffraction (XRD) patterns for these films are shown in Fig. 1c. The (001) diffraction peaks of n = 1 and 2 were found to have q = 0.37 Å⁻¹ and 0.29/0.30 Å⁻¹, respectively, while an (001) n = 3 peak could not be observed for these samples. The n = 2 peak experiences a small shift in the ⟨n⟩ = 3 film, possibly due to strain. These q values are similar for analogous films using the MA- and iodine-only composition $(VBABr)_2MA_{n-1}Pb_nI_{3n+1}$ (Supplementary Fig. 1), indicating that the RDPs with multication and multianion composition have minimal shifting of their characteristic (001) diffraction peaks. The diffraction peak positions we observe for n = 1 and 2 are also very similar to those we have previously measured and simulated for RDPs using the ligand phenethylammonium, PEA (owing to the structural similarity of PEA and VBA)[21], where we found n = 3 PEA RDPs to diffract at q = 0.22 Å. We assume a similar peak for the n = 3 VBA RDPs here.

We next used in situ GIWAXS in order to gain insight into the formation of the 2D/3D interface. Measurements were performed at the Brockhouse Beamline[22] at the Canadian Light Source using a custom homebuilt spinning platform (see Supporting Information for details of the experimental setup and execution). In situ GIWAXS patterns for films of MAPbI₃ treated with a solution of 5 mg/mL VBABr in isopropanol (IPA) are shown in Fig. 2a. Time

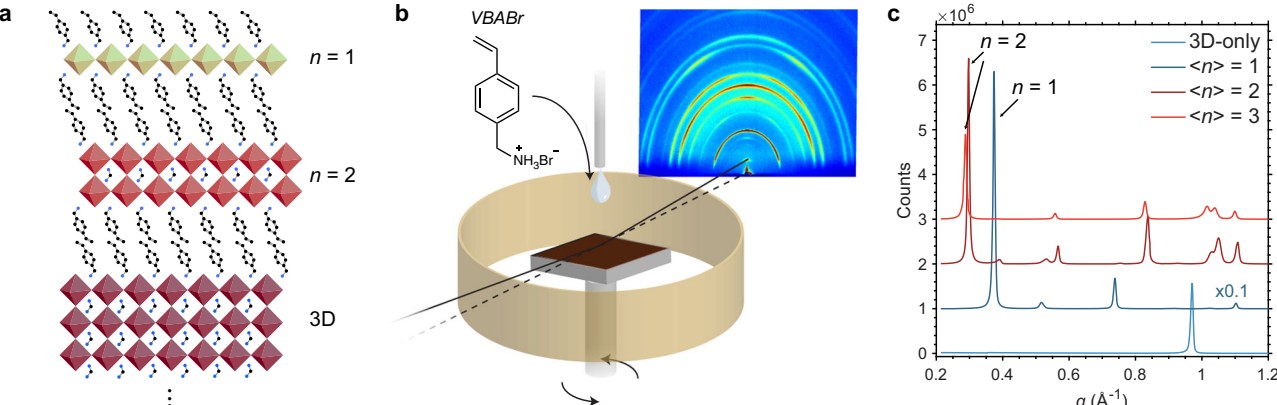

**Fig. 1 Diffraction from 2D/3D perovskite structures. a** Illustration of a 2D/3D interface for consecutive layers of 3D, n = 2, and n = 1 RDPs. **b** Schematic of the in situ GIWAXS experimental system used herein. A solvent dripper is positioned directly above the sample, which is mounted on a motorized pole with double-sided carbon tape. A Kapton shield encircles the spinning sample, prevented liquid from splashing onto the X-ray optics and the detector. VBABr: vinylbenzylammonium bromide. **c** Ex situ XRD patterns of RDP thin films with central n values of 1–3, and 3D control. Patterns are offset for clarity. Note: these are not 2D/3D perovskite thin films, and were synthesized from solutions containing precursor ratios corresponding to each n value.

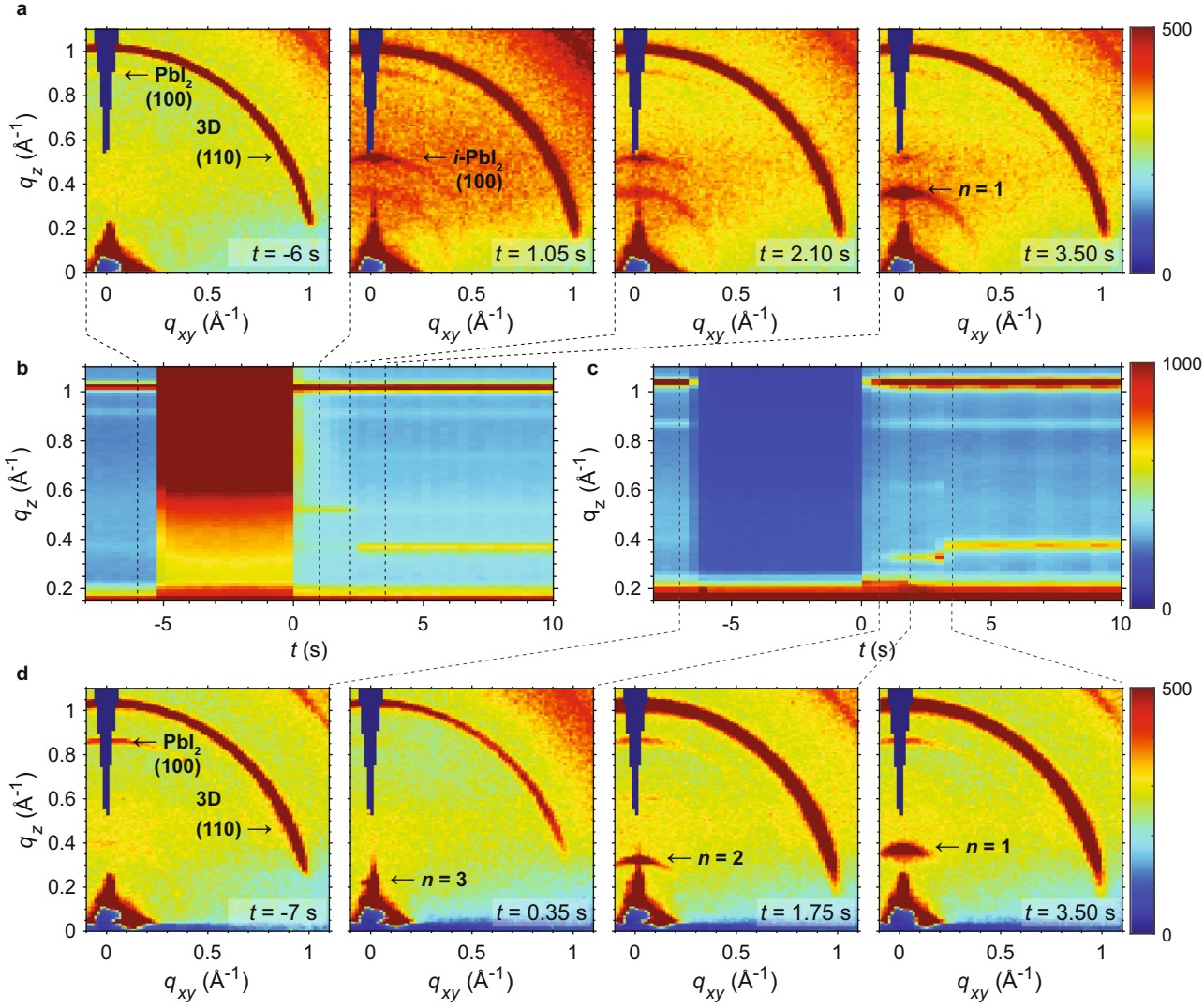

**Fig. 2 In situ GIWAXS patterns for (MAPbBr$_3$)$_{0.05}$(FAPbI$_3$)$_{0.95}$ and MAPbI$_3$. a** In situ GIWAXS patterns in the range of $q = -0.1$–$1.1$ Å$^{-1}$ for a film of MAPbI$_3$ treated with a solution of 5 mg/mL VBABr in IPA. $i$-PbI$_2$ = intercalated PbI$_2$. **b** Contour map of azimuthally integrated GIWAXS patterns for MAPbI$_3$. **c** Contour map and **d** in situ GIWAXS patterns for a film of (MAPbBr$_3$)$_{0.05}$(FAPbI$_3$)$_{0.95}$ treated with a solution of 1 mM VBABr in 3:97 IPA:CF. Patterns were collected with an incident angle of 0.1˚.

zero is taken to be the frame where spinning was engaged and the ligand solution is removed from the surface. In the initial frame ($t = -6$ s), we observe the expected isotropic diffraction from the (110) plane of 3D MAPbI$_3$ at $q \approx 1.0$ Å$^{-1}$, and also a smaller amplitude peak at $q_z \approx 0.9$ Å$^{-1}$, which corresponds to PbI$_2$. After soaking the surface with the ligand solution for ~6 s, the spinning is turned on at a speed of 1000 r.p.m. At $t = 1.05$ s after spinning is initiated, a peak at $q_z \approx 0.50$ Å$^{-1}$ is observed, which then ($t = 2.10$ s) decays simultaneously with growth of a peak at $q_z \approx 0.37$ Å$^{-1}$. This diffraction peak—which corresponds to the (100) plane of $n = 1$—then remains constant through to the end of the experiment, which can be seen from a contour map of azimuthally integrated GIWAXS patterns (Fig. 2b). The diffraction peaks for both the transient intermediate peak and the $n = 1$ peak are strongly centered along $q_{xy} = 0$, indicating that the species are oriented parallel with the substrate, which is often observed for low-$n$ RDPs[3,10,21,23].

These experiments indicate that $n = 1$ is not directly formed from the MAPbI$_3$ but rather transitions through an intermediate state. The peak at $q_z \approx 0.50$ Å corresponds to a $d$-spacing of 12.6 Å, which is too large to be attributed to PbI$_2$ and too small to

be attributed to an RDP ($n = 1$ has the smallest $d$ of ~17 Å, and higher $n$ RDPs have progressively larger $d$ as the thickness increases). Kim et al. observed a highly similar peak with $q \approx 0.55$ Å$^{-1}$ that was also oriented along $q_z$, during thermal degradation of MAPbI$_3$ into PbI$_2$ and other decomposition products[24]. This peak is attributed to 2D PbI$_2$ sheets that are intercalated by MAI or solvent molecules. We therefore posit that either IPA from the ligand solution or MAI that is being displaced from the perovskite lattice are the species that intercalate into PbI$_2$ sheets. This is supported by additional experiments using ligand solutions containing chlorobenzene (CB) with IPA:CB solvent ratios of 3:1, 1:1, and 1:3. When CB is included in the solvent, we do not observe the intermediate species, indicating that IPA intercalates with PbI$_2$ to form the intermediate species during 2D/3D interface formation. This suggests that these PbI$_2$ sheets template the subsequent formation of $n = 1$ RDPs, consistent with the fact the diffraction peaks for the intercalated PbI$_2$ and $n = 1$ are identical in shape and therefore orientation. Control experiments using only IPA as the exposure solution confirmed that this peak only appears when ligands are included (Supplementary Fig. 2), meaning cation displacement by the

ligand molecules is also required to form this intermediate intercalated $PbI_2$.

We observe a very different 2D/3D transformation from in situ GIWAXS experiments on multication and multianion perovskites of composition $(MAPbBr_3)_{0.05}(FAPbI_3)_{0.95}$. Compared to $MAPbI_3$, multication and -anion compositions achieve higher PCEs in photovoltaic devices[3,13,25]. Accordingly, in order to best simulate the 2D/3D interfaces used in our devices[10], we used a solution of 1 mM VBABr (~214.11 µg/mL) in 3:97 IPA:chloroform (CF). A lower proportion of IPA reduces perovskite dissolution and surface trap state density[13], and a lower concentration of ligands results in thinner RDP layers, which is favorable for carrier extraction and fill factor[11,12]. To compare fairly with our results from $MAPbI_3$, we also performed experiments with $(MAPbBr_3)_{0.05}(FAPbI_3)_{0.95}$ using 5 mg/mL VBABr in IPA, and observed no intermediate state before the formation of $n = 1$ (Supplementary Fig. 3).

A contour map of azimuthally integrated in situ GIWAXS patterns and four of the full patterns are shown in Fig. 2c and d. In contrast to $MAPbI_3$, rather than observing an intermediate at higher $q$, we instead observe a small but discernible peak that is overlapped with the specular reflection near $q_z \approx 0.22$ Å ($d \approx$ 28.6 Å). This peak is only visible for a single frame before disappearing simultaneously with the appearance of another peak at $q_z \approx 0.30$ Å ($d \approx 20.9$ Å), which remains for ~1 s before transforming into a peak at $q_z \approx 0.37$ Å ($d \approx 12.6$ Å). The three peaks are similar in shape and all oriented along $q_{xy} = 0$. These $d$-spacings correspond to (001) diffraction planes of $n = 3$, 2, and 1 RDPs.

In order to more clearly observe these transitions, we took the average of the first three frames of the experiment—which contained diffraction from the 3D perovskite, $PbI_2$, and the specular reflection—and subtracted them from all subsequent frames. A contour map of azimuthally integrated GIWAXS patterns and traces near time zero are shown in Fig. 3. We can see that the subtraction cleanly removes the specular reflection at low $q$ values, evidenced by first pattern in Fig. 3b which corresponds to the frame before the solvent exposure. Examining the patterns between $q_z = 0.15–0.45$ Å reveals the distinct peak at $q_z \approx 0.22$ Å in the earliest frames after spinning is initiated, and clearly shows the evolution of $n = 3 \rightarrow 2 \rightarrow 1$ in the first few seconds of spinning. The $n = 3 \rightarrow 2 \rightarrow 1$ transition is most apparent when using the 1 mM VBABr 3:97 IPA:CF solution, but a clear $n = 2 \rightarrow 1$ transition is also observed when using 5 mg/mL VBABr in a mixture 1:3 IPA:CB (Supplementary Fig. 4).

Our experiments thus suggest that for $(MAPbBr_3)_{0.05}(FAPbI_3)_{0.95}$ films, the transformation from 3D perovskites into RDPs occurs through a progressive reduction of dimensionality, rather than the immediate formation of $n = 1$. It is possible that even higher $n$ RDPs are formed initially, but the (001) diffraction peaks for these species would be at too low $q$ to be observed in our experiment. Moreover, in such a sequence of reducing dimensionality, we would expect diffraction of larger RDPs to be weaker, since there would initially be few instances of adjacent and periodic high-$n$ RDPs, but once they split into lower-$n$ RDPs, these structures would be immediately adjacent to each other, thus promoting diffraction[21]. Considering the small percentage of bromine in the $(MAPbBr_3)_{0.05}(FAPbI_3)_{0.95}$ films, we posit that the difference in the 2D/3D transformation pathways arises due to the greater stability of $FAPbI_3$ compared to $MAPbI_3$, where the latter is known to exhibit poorer thermal stability due to loss of volatile species under thermal stress and is more easily converted to $PbI_2$[26,27].

### DFT simulations of interface formation and stabilization.

In the multication films, we hypothesize that ligand molecules are able to infiltrate into and bisect a part of the 3D surface into high-$n$ domains, which undergo further bisection into lower-$n$ RDPs. We illustrate such a process in Fig. 4a. Further studies would be required to probe and confirm more microscopic picture of ligand infiltration and sequential bisection of the 3D lattice, where the terminating cations along the surface are replaced by the ligand molecule.

This idea is, however, supported by Density Functional Theory (DFT) calculations, where we calculated the binding energy of two perovskite fragments at their interfaces while their surfaces are converted from MAI-terminated to VBA-terminated (see Supporting Information for computational methods). These structures are shown in Fig. 4b, and a plot of the binding energy versus number of incorporated VBA molecules is shown in Fig. 4c. The binding energy increases, approximately linearly, with each VBA ligand substituted into the interface, from −0.10 eV for 1 VBA and up to −3.14 eV for 16 VBAs. If we gauge the reaction energies per VBA molecule insertion, the values are in −0.10, −0.11, −0.12, −0.14, and −0.20 eV for the systems with 1, 2, 4, 8, and 16 VBA molecules, respectively, showing that the thermodynamics become increasingly favorable due to the π–π stacking of the VBA molecules. We also examined binding energies at the interfaces between a fragment of $n = 1$ terminated

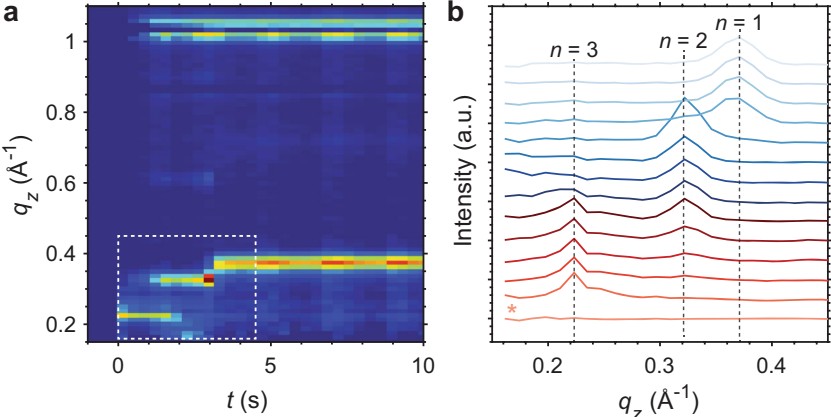

**Fig. 3 Background subtraction revealing $n = 3$. a** Contour maps of azimuthally integrated in situ GIWAXS patterns after subtracting the average of the first three frames before solvent exposure. **b** Patterns taken from within the dashed box indicated in panel **a**. The first pattern labelled with an asterisk is taken directly before the solvent exposure, confirming the effectiveness of subtracting the average of the first three frames.

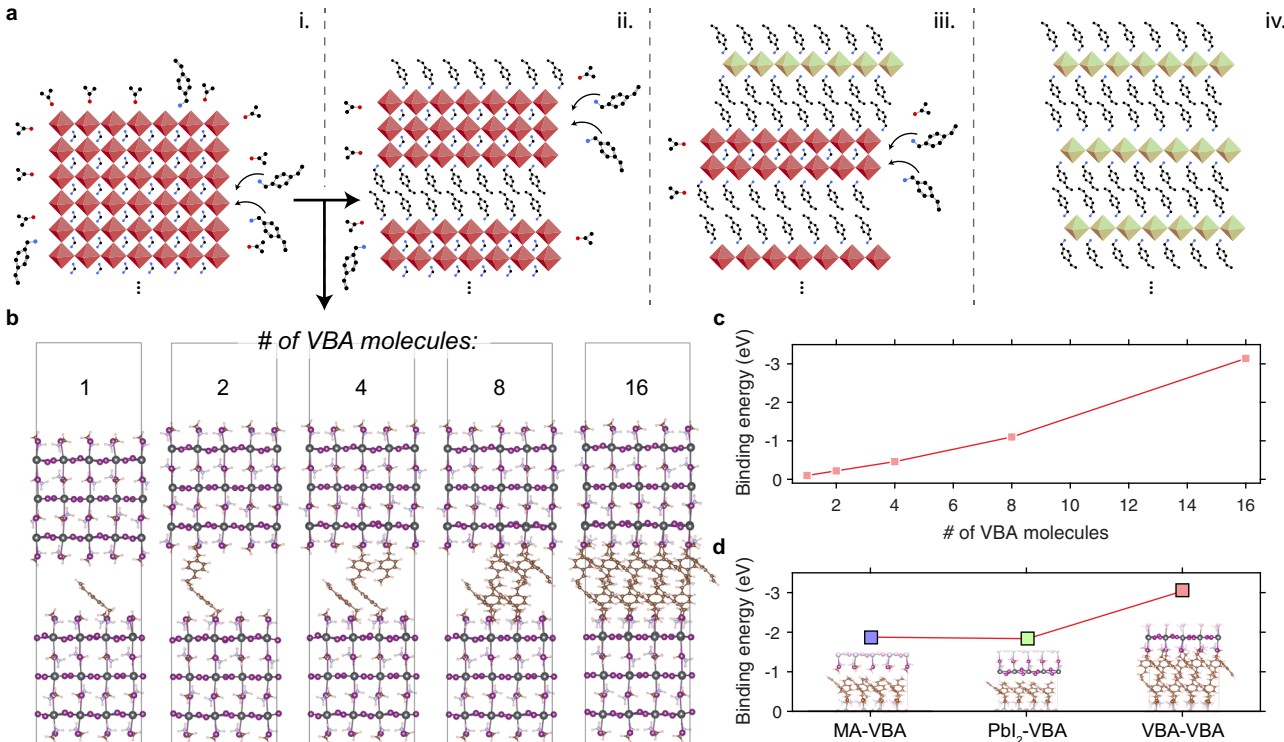

**Fig. 4 Proposed 3D-to-2D conversion mechanism and DFT simulations. a** Hypothesized transformation of a 3D perovskite surface into RDPs via (i) an initial infiltration of ligands and bisection of the lattice, (ii) formation of an $n = 3$ RDP and further bisection, (iii) formation of $n = 2$ and 1 RDPs, and (iv) final reduced-dimensional form of $n = 1$. **b** Models used in DFT calculations to calculate binding energy of adjacent fragments, where the number of VBA molecules that substitute MA molecules on the surface are indicated above each structure. **c** Binding energy versus number of VBA molecules and **d** binding energies at 2D/3D interfaces formed by a VBA-terminated $n = 1$ fragment sandwiched between 3D fragments terminated by MA (blue), $PbI_2$ (green), and VBA (red).

by VBA and a 3D fragment terminated by MA, $PbI_2$, and VBA (Fig. 4d), which we found to be −1.85, −1.84, and −3.06 eV, respectively, again demonstrating the favorable thermodynamics of VBA-VBA compared to other surface terminations of the 3D perovskite. This binding energy likely results from phenyl ring stacking: such interface stabilization via ligand interactions has been noted previously for RDPs using PEA ligands, and was invoked to explain the improved stability of low-$n$ RDPs compared to higher-$n$ and quasi-3D structures[4]. We additionally calculate binding energies considering more commonly used cations butylammonium (BTA) and PEA (Supplementary Fig. 5). The binding energies with 16 BTA and PEA molecules are −2.43 and −2.70 eV, respectively. The reaction energies per BTA and PEA insertion are −0.15 and −0.17 eV, respectively, compared to −0.20 eV for VBA. This supports our assertion that extended π–π stacking at the interface makes the interface growth more thermodynamically favorable. Here, we posit that the same ligand interactions are what cause the ligand molecules to bisect the 3D lattice and form an interface between two fragments, and thus drives 2D/3D interface formation.

**Influence of 2D/3D formation on photovoltaic efficiency.** Lastly, we examined the influence of the different ligand solutions on photovoltaic device performance. Devices were fabricated in an n-i-p architecture comprised of ITO/$SnO_2$/3D perovskite/ RDPs/Spiro-OMeTAD/Au, following refs. [3,10]. J–V curves for devices with 2D/3D active layers fabricated with different solvent conditions are shown in Fig. 5. Compared to control (3D-only) devices, the films exposed to 1 mM VBABr that undergo progressive dimensional reduction have higher fill factors (FF), open-

circuit voltages ($V_{OC}$), and overall higher PCEs. These improvements imparted by the 2D/3D interface, relative to 3D-only films, have been attributed to passivation by the RDPs and more favorable band alignment[3,9,10,12]. The 1 mM VBABr device also outperforms a series of devices exposed to 5 mg/mL VBABr in mixtures of IPA and CB (Fig. 5b). From this series, we see that decreasing the proportion of IPA leads to increased $V_{OC}$ and $J_{SC}$, whereas FF remains fairly constant throughout (besides for the pure IPA solution).

The increasing $J_{SC}$ with decreasing IPA proportion can partially be attributed to lessened dissolution of the underlying 3D layer when exposed to lower concentrations of IPA, which decreases the amount of 3D perovskite that is lost during the ligand exposure and preserves thicker, more absorptive active layers. This could also contribute to the trend in $V_{OC}$. However, we posit that the improved photovoltaic metrics are also sensitive to the assembly mechanism of the 2D/3D interface, which we have shown varies with different solvent combinations: when only IPA is used, we observed no intermediate peaks during in situ GIWAXS experiments, and $n = 1$ RDPs appear to form directly (Supplementary Fig. 3)—but when CB is included in the ligand solvent, we do observe the $n = 2 \rightarrow 1$ transition (Supplementary Fig. 4). IPA is the solvent which dissolves both VBABr and the perovskite, whereas the CB or CF antisolvent is included to help prevent dissolution of the perovskite surface. As the transient $n = 3$ and $n = 2$ peaks are only observed when CB or CF are included in the ligand solution, this suggests that the CB and CF antisolvent serves the additional function of slowing down the cation substitution reactions—which would require IPA to solvate MA, FA, and VBABr—that are involved with the overall 2D/3D transformation. This would be consistent with the $n = 3 \rightarrow 2 \rightarrow 1$

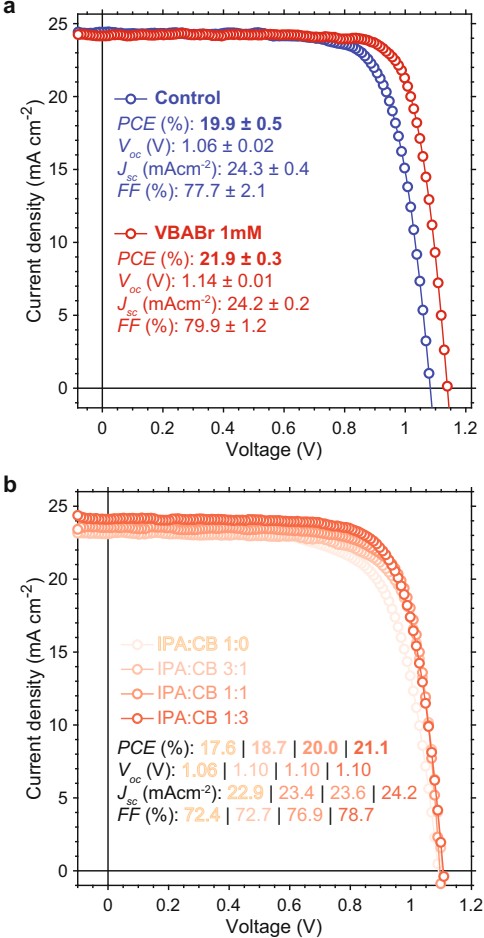

**Fig. 5 Photovoltaic devices for different 2D/3D formation conditions. a** Reverse scan J–V curves and device metrics for photovoltaics with active layers exposed to 1 mM VBABr in 3:97 IPA:CF and a control device that was not exposed to any ligand solution (3D-only), and for **b** active layers exposed to 5 mg/mL VBABr in different ratios of IPA and CB. Figures of merit shown here are taken from the average of multiple devices (standard deviations are shown alongside values in panel **a**, and are omitted in panel **b** for clarity. A boxplot of values measured for multiple devices and tabulated average values with standard deviations are given in Supplementary Fig. 6 and Supplementary Table 1). Forward scans were omitted here for clarity. See Supplementary Fig. 7 for forward and reverse scan J–V curves.

transition only being observed using a solution containing 1 mM VBABr and 3:97 IPA:CF as the solvent, meaning the 2D/3D transformation proceeds slowest under these conditions, whereas films exposed to 5 mg/mL VBABr in 1:3 IPA:CB undergo faster cation substitution reactions, explaining why only the $n = 2 \rightarrow 1$ transition is observed.

The combined observations from our in situ GIWAXS experiments and our devices suggest that slower 2D/3D transformations are correlated with improved device performance, since the 1 mM VBABr photovoltaic has the highest $V_{OC}$, FF, and PCE. This would also be consistent with the trend of increasing $V_{OC}$ and PCE with decreasing amounts of IPA for the solvent series. If we consider the 3D-to-2D transformation as analogous to crystal growth, then slower rates of transformation or crystallization could lead to higher quality RDPs and thinner 2D layers. This may improve the optoelectronic and morphological properties of the RDPs, as has been demonstrated generally

for bulk perovskite and RDP thin films used in photovoltaics and LEDs[28–30].

**2D/3D interface surface morphology and carrier extraction.** We collected atomic force microscopy (AFM) and scanning electron microscopy (SEM) images to examine the influence of the different solvent conditions and corresponding 2D/3D transformations on the film surface morphology (Fig. 6a, b). For films exposed to ligands dissolved in pure IPA, we observe large (micron-sized) unoriented flakes on the surface that are not conformal with the underlying 3D grains. Decreasing the concentration of the IPA reduces the amount of large unoriented crystallites and the morphology begins to resemble that of the 3D control film. For the lowest concentration of IPA using the 3:97 IPA:CF solution, the surface is highly similar to the 3D control (with slightly lower roughness), indicating that the RDPs are formed conformally on the surface. The 1:3 IPA:CB sample has the lowest surface roughness of the solvent series using 5 mg/mL VBABr solutions, and was also the only film in this series where we could observe an $n = 2 \rightarrow 1$ transition; whereas the 1 mM VBABr has an even lower surface roughness and was the only film to exhibit the $n = 3 \rightarrow 2 \rightarrow 1$ transitions. These observations are consistent with our hypothesis that lowering the concentration of IPA can slow down the growth of RDPs during 2D/3D interface formation: high concentrations of IPA causes rapid growth of large RDP grains that are unoriented relative to the 3D surface, whereas high concentrations of antisolvent instead facilitate conformal growth of the RDPs on the 3D grains. The slowest reaction rates may therefore yield higher quality RDPs through improved passivation and better surface conformity, which would explain the improved device performance.

We performed Time-of-Flight Secondary Ion Mass Spectrometry (ToF-SIMS) to estimate the thickness of the RDP layers using depth profiles of the ligand-derived fragment $C_9H_{12}N^+$ (Supplementary Fig. 10). We roughly estimate RDP layer thicknesses of 55, 40, 30, and 35 nm for IPA:CB ratios of 1:0, 3:1, 1:1, and 1:3, respectively, and 4 nm for the 1 mM solution. This is indicative of the faster RDP growth rates forming thicker layers, which is known to hinder charge extraction from 2D/3D interfaces[12].

To understand how the different 2D/3D interfaces affect charge extraction in photovoltaic devices, we performed photoluminescence (PL) lifetime quenching experiments by comparing lifetimes with and without a Spiro-OMeTAD quencher layer atop the 2D/3D interface (Fig. 6c)[31–33]. Aside from the 3D film, biexponential fits were required to fit the PL lifetimes. The values shown correspond to the longer of the two fitted lifetimes, which is most relevant for evaluating the extent of diffusion to and extraction at the quencher layer. The 3D film shows the strongest quenching, and the 1 mM VBABr film is also significantly quenched (and has the longest unquenched lifetime). For the remaining films, we observe that 2D/3D interfaces formed with higher concentrations of IPA have lifetimes that are even longer in the presence of the quencher, indicative of poor extraction of holes by Spiro-OMeTAD. This is consistent with our results from photovoltaic devices, where higher IPA concentrations lead to lower fill factors.

We have demonstrated using in situ GIWAXS that the formation of 2D/3D interfaces proceeds via intermediate states that differ depending on perovskite composition. For MAPbI₃, exposure to the ligand solution induces the formation of a 2D layer of $PbI_2$ intercalated by solvent molecules, which is subsequently converted into $n = 1$ RDPs. For $(MAPbBr_3)_{0.05}(FAPbI_3)_{0.95}$ films, there is a progressive dimensional reduction from 3D fragments into $n = 3 \rightarrow 2 \rightarrow 1$. DFT calculations find

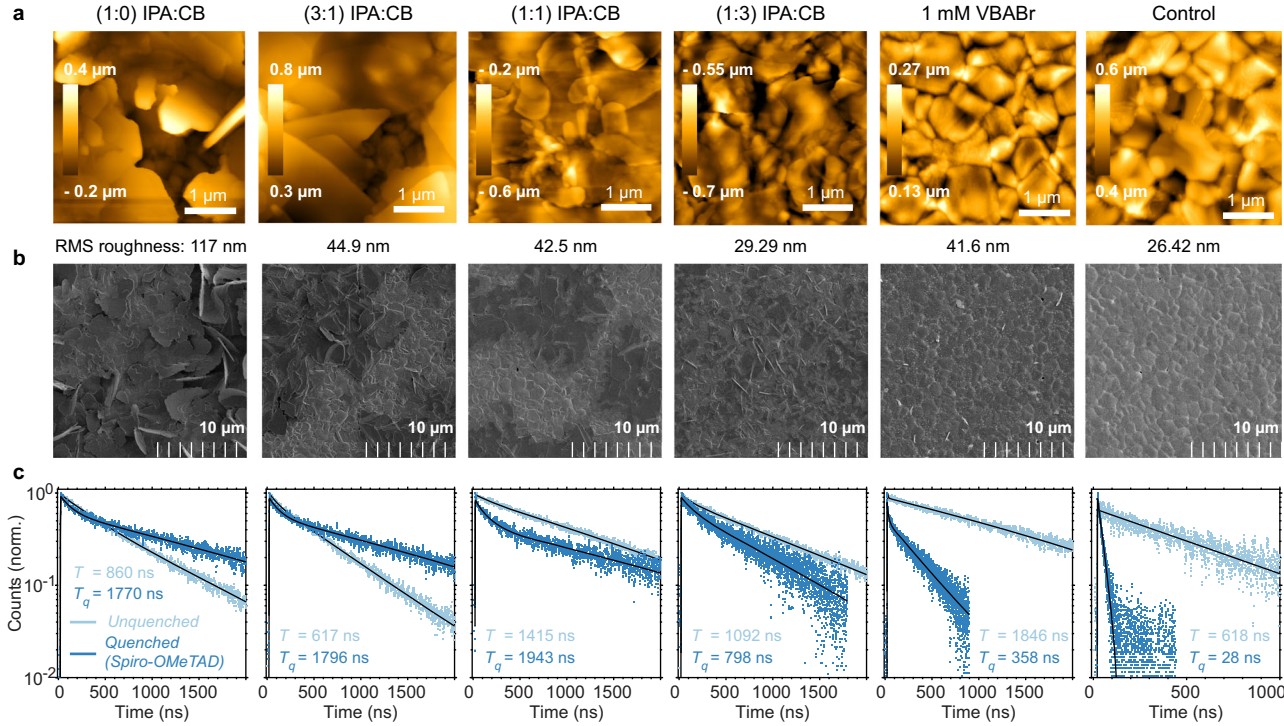

**Fig. 6 2D/3D surface morphology and carrier extraction. a** AFM height images with root mean square (RMS) roughness and **b** SEM images of films treated with different ligand solutions. **c** PL lifetimes and fits for the same films with ($T_q$, quenched) and without ($T$, unquenched) a Spiro-OMeTAD layer spin-coated on top of the 2D/3D interfaces. The films were photoexcited (532 nm), and emission collected, on the substrate side of the film (glass/3D perovskite interface). AFM and SEM images at different ranges and magnifications are shown in Supplementary Figs. 8 and 9, and fitted PL lifetime parameters are shown in Supplementary Table 2.

that the binding energy at the interface between two perovskite fragments increases upon substitution with VBA molecules, owing to the ring stacking interactions afforded by these ligand cations. We use these observations to hypothesize that the ligands infiltrate and bisect the 3D perovskite to form lower-dimensional fragments that continuously split into lower-$n$ RDPs before becoming $n = 1$, in accordance with our observations from in situ GIWAXS experiments. By comparing photovoltaic devices with 2D/3D interfaces fabricated using different ligand solutions, we posit that lowering the concentration of IPA in the ligand solution slows the interface formation, which leads to improved photovoltaic device performance. This is supported by AFM, SEM, and ToF-SIMS measurements that reveal thicker and more disordered RDP layers are formed with higher concentrations of IPA, which leads to poorer carrier extraction from 2D/3D interfaces as determined using PL lifetime quenching. Our work highlights how a fundamental understanding of 2D/3D interface formation can inform on the design of superior devices. Future studies will be required to definitively prove these microscopic mechanisms we propose based on our experimental and computational observations.

## Methods

**Synthesis of 4-vinylbenzylammonium bromide**. One gram of 4-vinylbenzylamine (VBA, 0.98 mL, 7.36 mmol, stored in a freezer prior to use) was added to 15–30 mL of cold isopropanol (cooled in an ice bath). 0.65 mL of a solution of 48% HBr (6.62 mmol) was added dropwise to the cooled VBA/isopropanol mixture. This mixture was left to react for 2 h, slowly warming up to room temperature. The solvent was then evaporated under reduced pressure and the remaining white solid was left to dry for an additional 30 min under mild heating and vacuum (temperature was maintained at a maximum of 70 °C). The solid was then stirred in ~20 mL diethyl ether for 10–15 min before being filtered and washed five times with diethyl ether. This washed solid white powder was then recrystallized by dissolving in boiling isopropanol and subsequently re-precipitated with diethyl ether. The recrystallized solid was isolated via vacuum filtration,

washed an additional three times with diethyl ether, and then held under vacuum for 16 h before being brought into a nitrogen glovebox for storage and use. We performed [1]H NMR on the washed and recrystallized products to ensure that the vinyl groups remained unreacted after exposure to heat when drying and recrystallizing the solid.

**Formation of 2D/3D perovskites on top surface**. For 5 mg/mL ligand solutions, 5 mg of VBABr was dissolved in 1 mL of mixtures of isopropanol and chlorobenzene. For 1 mM (~214.11 μg/mL) films, we first prepare a 10 mM stock solution in a 1:10 mixture of isopropanol and chloroform, before diluting with chloroform to achieve a VBABr concentration of 1 mM. Following the addition of chloroform, the solution will appear cloudy; adding a few drops (~20 μL) of isopropanol and vigorous shaking clarifies the solution. In all cases, the ligand solution is passed through a 0.22 μm PTFE filter before use. For devices, 200–300 μL of ligand solution is spin-coated onto annealed 3D films at a spin speed of 4000 r.p.m. for 20 s. The films are then transferred onto a hot plate to anneal at 150 °C for 30 s. This procedure follows refs. [3,10].

**Deposition of $\langle n \rangle = $ 1, 2, 3 RDPs**. PbI$_2$, PbBr$_2$, MAI, MABr, and FAI precursors were dissolved according to formulas (VBABr)$_2$MA$_{n-1}$Pb$_n$I$_{3n+1}$ and (VBABr)$_2$(FA$_{0.95}$MA$_{0.05}$)$_{n-1}$Pb$_n$(I$_{0.95}$Br$_{0.05}$)$_{3n+1}$ in 4:1 DMF:DMSO with [Pb$^{2+}$] = 0.8 M. The solutions were stirred at 70 °C for 2 h, cooled to room temperature, filtered (0.22 μm, PTFE), and spin-coated on cleaned glass slides at 4000 r.p.m. for 20 s. Films are annealed at 70 °C for 10 min.

**X-ray diffraction**. XRD patterns were obtained using a Rigaku MiniFlex 600 diffractometer (Bragg-Brentano geometry) equipped with a NaI scintillation counter detector and a monochromatized Cu Kα radiation source ($\lambda = 1.5406$ Å) operating at a voltage of 40 kV and current of 15 mA.

**In situ GIWAXS**. GIWAXS measurements were conducted at the Hard X-ray MicroAnalysis (HXMA) and Brockhouse X-ray Diffraction and Scattering Beamlines - Low Energy Wiggler (BXDS-WLE) beamlines of the Canadian Light Source (CLS). An energy of 17.998 keV ($\lambda = 0.6888$ Å) was selected using a Si(111) monochromator. Patterns were collected on a SX165 CCD camera (Rayonix) placed at a distance of 157 mm from the sample. A lead beamstop was used to block the direct beam. Images were calibrated using LaB6 and processed via the Nika software package (ref. [34]) and the GIXSGUI MATLAB plug-in (ref. [35]). We built a sample holder with a motorized pole capable of rotating at 500–5000 r.p.m.

and integrated the unit into the fully translatable and rotatable GIWAXS sample stage. This enabled us to alter the incident angle of the X-rays and the height of the spinning sample. After mounting a 3D perovskite sample, 100–200 μl of the ligand solution was drawn into tubing directly above the sample. We then began collecting GIWAXS patterns every 0.35 s (limited by detector readout time). A few frames are collected for the bare 3D perovskite before the solvent is released. After waiting ~5–7 s for the solvent to soak the surface, the motor is turned on and the sample begins to spin at 500–2000 r.p.m. (the limit of the motor). Patterns were collected for up to 30–60 s, depending on the experiment. We found that slower spin speeds were required to observe transient species discussed herein, which otherwise were not captured with our time resolution of 0.35 s. Comparing ex situ GIWAXS experiments performed over a range of 500–2000 r.p.m., we found that slower spin speeds resulted in stronger diffraction from the 2D perovskites, but otherwise qualitatively similar patterns (see Supporting Information, Supplementary Fig. 11).

**Computational details**. First-principles calculations based on density functional theory (DFT) are carried out as implemented in the PWSCF Quantum-Espresso package[36]. Geometry optimization, including dispersion correction[37], is performed using GGA-PBE[38] level of theory and the electrons-ions interactions are described by ultrasoft pseudo-potentials[39] with electrons from I 5s, 5p; Br 4s, 4p; N, C 2s, 2p; H 1s; Pb, 6s, 6p, 5d; shells explicitly included in calculations. Plane-wave basis set cutoffs for the smooth part of the wave functions and the augmented density were 40 and 320 Ry, respectively. Geometry optimizations are done with Gamma kpoint sampling for $2 \times 2$ supercell slabs using the experimental cell parameters of tetragonal MAPbI$_3$ along the periodic direction of the slabs, whereas a vacuum of more than 10 Å is introduced along the surface truncation direction. The binding energies in Fig. 4b are calculated using the following formula:

$$\text{Binding energy} = [E_{total} - E_{fragment1} - E_{fragment2}]$$

where $E_{total}$ represents the relaxed energy of the entire systems, whereas $E_{fragment1}$ and $E_{fragment2}$ represent the single-point energy of the fragments cut from the optimized models. The binding energies for Fig. 4d are calculated using the following formula:

$$\text{Binding energy} = [E_{2D/3D} - E_{3D-fragment1} - E_{3D-fragment2} - E_{2D-fragment}]/2$$

where $E_{2D/3D}$ represents the relaxed energy of the 2D/3D systems, whereas $E_{3D-fragment1}$, $E_{3D-fragment2}$, and $E_{2D-fragment}$ represent the single-point energy of the fragments cut from the optimized 2D/3D slabs.

**Device fabrication**. Devices were fabricated following identical procedures as in ref. [2]. ITO glass was cleaned by sequentially washing with detergent, deionized water, acetone, and isopropanol (IPA). ITO was then cleaned with ultraviolet ozone for 15 min. The substrate was then spin-coated with a thin layer of SnO$_2$ nanoparticle solution (1:3:3, SnO$_2$ (15% in water):IPA:water) at 3000 r.p.m. for 20 s, and annealed in ambient air at 150 °C for 1 h. To fabricate perovskite solar cells based on (FAPbI$_3$)$_{0.95}$(MAPbBr$_3$)$_{0.05}$, the perovskite solution was prepared by dissolving 889 mg ml$^{-1}$ of FAPbI$_3$, 33 mg ml$^{-1}$ of MAPbBr$_3$, and 33 mg ml$^{-1}$ of MACl in DMF/DMSO (8:1 v/v) mixed solvent. Then, the solution was coated onto the ITO/SnO$_2$ substrate by two consecutive spin-coating steps, at 1000 and 2500 r.p.m. for 5 and 20 s, respectively. During the second spin-coating (2500 r.p.m.) step, 1 ml of diethyl ether was poured onto the substrate after 15 s. The intermediate phase substrate was then put on a hot plate at 150 °C for 10 min. After the fabrication of the 3D perovskite film, the 2D layer was fabricated by depositing a solution of 4-vinylbenzylammonium bromide in mixed solvent (5 mg/mL in 1:0, 3:1, 1:1, and 1:3 IPA:CB or 1 mM in 3:97 IPA:CF) onto the perovskite film, for 3–5 s, and then spinning the substrate at 4000 r.p.m. for 10 s. Then, the substrate was heat-treated at 150 °C for 30 s. For deposition of the hole-transport material, a spiro-OMeTAD solution in chlorobenzene (CB) (90.9 mg ml$^{-1}$) was prepared, and 23 μl of lithiumbis(trifluoromethanesulfonyl) imide (Li-TFSI) solution in acetonitrile (ACN) (540 mg ml$^{-1}$) and 39 μl of pure 4-tert-butylpyridine (tBP) were added in 1.1 ml of the solution. The spiro-OMeTAD solution including additives was spin-coated onto the perovskite surface at 1750 r.p.m. for 30 s. Finally, a gold electrode was deposited by thermal evaporation.

**Device testing**. The current density-voltage (J–V) characteristics were measured using a Keithley 2400 source meter under illumination from a solar simulator (Newport, Class A) with a light intensity of 100 mW cm$^{-2}$ (checked with a calibrated reference solar cell from Newport). Unless otherwise stated, the J–V curves were all measured in a nitrogen atmosphere with a scanning rate of 100 mV s$^{-1}$ (voltage step of 20 mV and delay time of 200 ms). The active area of 0.049 cm$^2$ was defined by the aperture shade mask placed in front of the solar cell. A spectral mismatch factor of 1 was used for all J–V measurements. The chamber containing the cells for testing was purged with and maintained under nitrogen flow during the measurements.

**Atomic force microscopy**. AFM images were collected using a Cypher in tapping mode with a fast-scanning silicon tip (calibrated spring constant $k = 10.6$ N/m, $f = 1393.303$ kHz, tip model: FS-1500AuD from Asylum Research).

**Scanning electron microscopy imaging**. SEM imaging was done using a Hitachi SU 5000, Schottky Field Emission, scanning electron microscope, at 1.0 kV beam energy.

**Time-of-flight secondary ion mass spectrometry (ToF-SIMS)**. The samples were examined using an ION-TOF (GmbH) TOF-SIMS IV equipped with a Bi cluster liquid metal ion source. A pulsed 25 keV Bi$_3^+$ cluster primary ion beam was used to bombard the sample surface to generate secondary ions. The negative secondary ions were extracted from the sample surface, mass separated, and detected via a reflectron-type of time-of-flight analyzer, allowing parallel detection of ion fragments having a mass/charge ratio (m/z) up to ~900 within each cycle (100 μs). A pulsed, low energy electron flood was used to neutralize sample charging. This technique is extremely surface sensitive, probing only the top 1–3 nm of the sample. The detection limits are believed to be in the range of ppb–ppm, depending upon the ion yield of different elements or species. Note that ToF-SIMS is not a quantitative analytical technique because ion yields for different elements are very different and dependent on the chemical environment in which the elements exist (matrix effect).

In order to depth profile the five samples, a 10 keV C$_{60}^+$ ion beam was used to sputter the surface in an area of 200 μm × 200 μm, and positive ion mass spectra were collected at 128 × 128 pixels over a smaller area (128 μm × 128 μm) within the sputtered area. The depth profile data were obtained by sputtering the surface with the C$_{60}^+$ beam for 2 s followed by a 0.5-s pause before Bi$_3^+$ was used to analyse the newly-generated surface. The depth of the crater generated as a result of the depth profiling was measured using a mechanical stylus profilometer (KLA-Tencor P-17), which allows the calibration of depth.

**Photoluminescence lifetime quenching**. Perovskite films were deposited onto cleaned glass slides, followed by a Spiro-OMeTAD layer (both layers were coated under the same conditions as used for devices, see above). PL lifetime measurements were performed using a Horiba Fluorolog Time Correlated Single Photon Counting (TCSPC) system with photomultiplier tube detectors. A pulsed laser diode (504 nm, 110–140 ps pulse width) was used as the excitation source with a 400–1600 ns period (0.28 nJ per pulse) was used to capture accurate lifetimes carrier lifetimes. Excitation and emission collection were performed on the bottom (glass) side of the film.

**Reporting summary**. Further information on experimental design is available in the Nature Research Reporting Summary linked to this paper.

## Data availability
All the datasets that support the findings of this study are available from the corresponding author upon reasonable request.

## Code availability
All code used to analyze data and support the findings of this study are available from the corresponding author upon reasonable request. GIWAXS patterns were analyzed using GIXSGUI in MATLAB (https://www.aps.anl.gov/Science/Scientific-Software/GIXSGUI)[35].

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

## Acknowledgements

This work was supported in part by the US Department of the Navy, Office of Naval Research (grant awards no. N00014-17-1-2524 and N00014-20-1-2572), the Natural Sciences and Engineering Research Council of Canada (NSERC), and the Canada Foundation for Innovation (CFI). A.H.P. was supported by the Canada Graduate Scholarships program from NSERC. The authors thank the Canadian Light Source (CLS) for support in the form of a travel grant. GIWAXS patterns were collected at the BXDS-WLE Beamline at the CLS with the assistance of Dr. Chang-Yong Kim and Dr. Adam Leontowich. The CLS is funded by NSERC, the Canadian Institutes of Health Research, CFI, the Government of Saskatchewan, Western Economic Diversification Canada, and the University of Saskatchewan. S.E.M. images were taken at the Ontario Center for the Characterization of Advanced Materials (OCCAM). ToF-SIMS measurements were performed at Surface Science Western at the University of Western Ontario.

## Author contributions

A.H.P., A.J., S.T., and R.Q.-B. conceived the idea, designed the experiments, and performed in situ GIWAXS experiments. A.J and S.T. performed XRD and PL quenching experiments. S.T. and L.G. fabricated films and solar cell devices. A.M. and F.D.A. performed Density Functional Theory simulations. E.H.J. advised on solar cell fabrication. T.C. and T.F. performed AFM measurements. C.-Y.K. provided technical assistance and supervision for GIWAXS experiments at the CLS. F.D.A. and S.O.K. provided technical feedback for the manuscript. A.H.P., A.J., S.T., A.M., and E.H.S. wrote the manuscript. All authors read and commented on the manuscript.

## Competing interests

The authors declare no competing interests.
