## [Peer Review File · Nature Communications]

Reviewers' comments:

Reviewer #1 (Remarks to the Author):

The manuscript describes a detailed study of the formation of 2D top layers on exposing 3D perovskite materials to a solution of ligands normally found in 2D perovskites. This is a very relevant subject as the introduction of 2d/3d interfaces strongly enhances the stability of devices based on these. In this sense the manuscript offers interesting new insights, although the argumentation in the paper is not always very convincing.

The GWAX measurements clearly show formation of 2D material, with possible intermediate structures depending on the solvent conditions. But the characterization and mechanisms are not so well supported.

- In the assignment of the diffraction peaks the authors rely on peak positions for other pure 2D materials (PEA), which is unfortunate.
- The DFT calculations used to support the claims are interesting but in order to say something about structure and (average) binding energies it would be necessary to have some idea of the dynamics, rather than just energetics based on geometry optimisations (i.e. 0 K). Dynamics in such materials is substantial and strong dependent on the nature of the organic molecule used.
- For the trends discussed in the device performance it would be interesting to know something about the statistical relevance. Are these trends based on a single device? What are the variations in PCE, FF, etc. if multiple devices are made in the same approach and are the reported differences larger than these variations?
- There seems to be differences between pure MAPI and the mixed cation material, but the authors do not really explain these differences. Is there any reason for the differences?
- A final point: a large part of the manuscript deals with very technical aspects of spin speeds etc and this obscures the story line quite a bit part of this technical could easily be summarised more briefly, resulting in a clearer story.

I therefore recommend major revisions.

Reviewer #2 (Remarks to the Author):

In this work Proppe et al have studied the crystallization dynamic of a specific 3D/2D perovskite interfaces revealing the steps happening during crystallization. Here the authors have used GIWAX to resolve in situ the 2D crystallization dynamics.

As the authors mentioned, there is a large variability and still no defined understanding on the 3D/2D crystallization dynamics which have an important impact on 3D/2D perovskite device optimization.

However, this is intimately due to the method used to incorporate the 2D perovskites such as direct cation incorporation in the solution, two step deposition in IPA, whereas the author mentioned the 3D surface is exposed to a solution containing a low concentration of the 2D cation which transform part of the surface into RDPs, or in the antisolvent. In addition, the choice of the 2D cation influences the resulting surface. For the specific method (or specific cation used), several groups already reported in situ dynamics through GIWAX (see for instance *Advanced Functional Materials* 31 (4), 2007473; *Chem. Mater.* 2019, 31, 22, 9472–9479), showing a very similar concept. Crystal formation has been already detailed in this work:

<https://doi.org/10.1002/adfm.202001294>.

As a result, this work, despite of the interest for the community, deals with 1. a specific method; 2. a specific cation for 2D which is not part of the commonly used cations and 3. a method which has been already proven.

For these reasons there is not justified novelty nor general interest for publication in *Nat. Comm.* I would rather suggest a minor, more specialized Journal. In addition, the authors should provide at

least a study case also for commonly used cations such as PEAI or BAI. Importantly, the authors should also provide device statistics and stability data using consensus aging tests.

Reviewer #3 (Remarks to the Author):

The authors report a study on reduced dimensional perovskites (RDPs) on top of 3D perovskite thin films by performing in-situ GIWAXS. In short, they find that different rate of RDP formation and pathways (either formation of an intermediate phase or higher n layers first before eventually forming n=1 layers) depending on the 3D perovskite thin film composition and solvent choices. They correlate higher solar cell performance with slower formation rate of RDPs. Based on the results and discussion presented in the manuscript, I cannot recommend publication for the following reasons:

1. It is not clear the solar cell results are statistically significant as there is no information on the number of devices and the distribution of relevant metrics.
2. With the complex formation pathways and multiple factors changing together with different ligand solution combinations, it is unclear whether the relationship between slower formation rate of RDPs and higher solar cell performance can be established. For example, other than GIWAXS, there is no other structural characterization of the thin film. We do not know whether the treatment changes the grain size and etc. Moreover, other than solar cell results (which are convolution of many factors), there is no other electrical and optical characterization that support their claims.

Based on these reasons, I'm not convinced by authors' claims and cannot say I obtained much meaningful new insights. Therefore, I do not recommend publication.

Reviewers' comments:

Reviewer #1 (Remarks to the Author):

The manuscript describes a detailed study of the formation of 2D top layers on exposing 3D perovskite materials to a solution of ligands normally found in 2D perovskites. This is a very relevant subject as the introduction of 2d/3d interfaces strongly enhances the stability of devices based on these. In this sense the manuscript offers interesting new insights, although the argumentation in the paper is not always very convincing.

The GWAX measurements clearly show formation of 2D material, with possible intermediate structures depending on the solvent conditions. But the characterization and mechanisms are not so well supported.

- In the assignment of the diffraction peaks the authors rely on peak positions for other pure 2D materials (PEA), which is unfortunate.

We only use the XRD patterns of single crystals of PEA materials to identify the peak position of $n = 3$. The other peak positions discussed here are taken directly from XRD patterns of pure $n = 1$ and nearly pure $n = 2$ thin films of VBA-based RDPs. We show below a plot of XRD patterns for $n = 1, 2,$ and 3 VBA thin films synthesized directly from precursors in solution, and for $n = 1, 2,$ and 3 PEA single crystal powders. The peak positions for $n = 1$ and 2 are the same for both ligand cations, as is to be expected considering their similar structure and size. The $n = 3$ peak is absent from the VBA films because dimensional mixing during thin film formation prevents clear diffraction.¹ It is likely that the vinyl group of VBA would react at the higher temperatures, which would prevent us from acquiring single crystals.

Considering the very similar peak positions of $n = 1$ and 2 for VBA and PEA, we feel it is fair to assume a similar position for $n = 3$.

- The DFT calculations used to support the claims are interesting but in order to say something about structure and (average) binding energies it would be necessary to have some idea of the dynamics, rather than just energetics based on geometry optimisations (i.e. 0 K). Dynamics in such materials is substantial and strong dependent on the nature of the organic molecule used.

Generally, *ab initio* molecular dynamics add the contribution of thermal energy and entropic effect to its static counterpart. It has been previously reported that PEA cations' rotational dynamics do not vary significantly with temperature due to the strong π - π stacking.²

However, the motivation of our calculations was to show how progressive organic cation stacking leads to the interface growth to a thermodynamically favorable process, which was well reflected by our calculations. If we gauge the reaction energies per VBA molecule insertion, the values results in -0.10, -0.11, -0.12, -0.14 and -0.20 eV for the systems with 1, 2, 4, 8 and 16 VBA molecules, respectively, which shows that the thermodynamics become more and more favorable due to the π - π stacking of the VBA molecules.

Additionally, to quantify such π - π stacking, we have now calculated the binding energy with commonly used cations butylammonium (BTA) and phenethylammonium (PEA). The binding energy with 16 BTA and PEA molecules systems are -2.43 and -2.70 eV, respectively. Thus the reaction energy per BTA and PEA insertion results in -0.15 and -0.17 eV, respectively, whereas the value is -0.20 eV for VBA. The presence of more π - π stacking at the interface makes the interface growth more thermodynamically favorable. Nevertheless, we agree that a complete dynamical dataset of intermediates would be comprehensive information for the reader, but that is beyond the scope of this manuscript and will be the subject of future work.

Figure S5. DFT models used to calculate the binding energies of BTA-BTA and PEA-PEA interfaces.

The following text has been added to page. 10 of the manuscript:

“If we gauge the reaction energies per VBA molecule insertion, the values are in -0.10, -0.11, -0.12, -0.14 and -0.20 eV for the systems with 1, 2, 4, 8 and 16 VBA molecules, respectively, which showing that the thermodynamics becomes more and more favorable due to the π - π stacking of the VBA molecules.”

“We additionally calculate the binding energy considering more commonly used cations butylammonium (BTA) and PEA (Fig. S5). The binding energies with 16 BTA and PEA molecules systems are -2.43 and -2.70 eV, respectively. The reaction energies per BTA and PEA insertion are -0.15 and -0.17 eV, respectively, compared to -0.20 eV for VBA. This supports our assertion that extended π - π stacking at the interface makes the interface growth more thermodynamically favorable.”

- For the trends discussed in the device performance it would be interesting to know something about the statistical relevance. Are these trends based on a single device? What are the variations in PCE, FF, etc. if multiple devices are made in the same approach and are the reported differences larger than these variations?

We now include device statistics from multiple devices, and find that the trends are consistent across the batch tested. We have included this new data in Figure S6 and Table S1 of the Supporting Information (they are also shown directly below this response for convenience). We have updated the values shown in Figure 4 of the main text to reflect the average values instead. The standard deviations for the different figures of merit are small (<1% in most cases).

Fig. S6. Boxplots of V_{oc} , J_{sc} , FF, and PCE for different film treatment conditions. A tabulation of the different values and uncertainties can be found below (Table S1).

	Control	IPA:CB (1:0)	IPA:CB (3:1)	IPA:CB (1:1)	IPA:CB (1:3)	IPA:CF (3:97)
V_{oc} (V)	1.06 ± 0.02	1.06 ± 0.07	1.10 ± 0.01	1.10 ± 0.01	1.10 ± 0.02	1.14 ± 0.01
J_{sc} (mA cm⁻²)	24.25 ± 0.40	22.87 ± 0.56	23.36 ± 0.17	23.62 ± 0.15	24.23 ± 0.15	24.16 ± 0.17
PCE (%)	19.93 ± 0.54	17.60 ± 1.46	18.73 ± 0.69	19.96 ± 0.37	21.06 ± 0.25	21.90 ± 0.28
FF (%)	77.67 ± 2.13	72.35 ± 2.17	72.65 ± 2.09	76.92 ± 2.25	78.70 ± 0.81	79.88 ± 1.16

- There seems to be differences between pure MAPI and the mixed cation material, but the authors do not really explain these differences. Is there any reason for the differences?

We posit that the differences in the 2D/3D transformation for MAPbI₃ and (MAPbBr₃)_{0.05}(FAPbI₃)_{0.95} are related to the poorer overall stability of MAPbI₃. MAPbI₃ more easily (compared to FAPbI₃) undergoes thermal decomposition into PbI₂ due to loss of volatile species such as HI and CH₃NH₂. We believe that this more favourable decomposition pathway for MAPbI₃ is what leads to the initial formation of PbI₂ before being converted into *n* = 1 RDPs. We have included the following texts and references on page 9 of the manuscript:

“Considering the small percentage of bromine in the (MAPbBr₃)_{0.05}(FAPbI₃)_{0.95} films, we posit that the difference in the 2D/3D transformation pathways arises due the greater stability of FAPbI₃ compared to MAPbI₃, where the latter is known to exhibit poorer thermal stability due to loss of volatile species and is more easily converted to PbI₂.^{3,4}”

- A final point: a large part of the manuscript deals with very technical aspects of spin speeds etc and this obscures the story line quite a bit part of this technical could easily be summarised more briefly, resulting in a clearer story.

We thank the author for their suggestion and have moved these technical details from the manuscript into the Supporting information.

I therefore recommend major revisions.

Reviewer #2 (Remarks to the Author):

In this work Proppe et al have studied the crystallization dynamic of a specific 3D/2D perovskite interfaces revealing the steps happening during crystallization. Here the authors have used GIWAX to resolve in situ the 2D crystallization dynamics.

As the authors mentioned, there is a large variability and still no defined understanding on the 3D/2D crystallization dynamics which have an important impact on 3D/2D perovskite device optimization.

However, this is intimately due to the method used to incorporate the 2D perovskites such as direct cation incorporation in the solution, two step deposition in IPA, where as the author mentioned the 3D surface is exposed to a solution containing a low concentration of the 2D cation which transform part of the surface into RDPs, or in the antisolvent. In addition, the choice of the 2D cation influences the resulting surface. For the specific method (or specific cation used), several groups already reported in situ dynamics through GIWAX (see for instance *Advanced Functional Materials* 31 (4), 2007473; *Chem. Mater.* 2019, 31, 22, 9472–9479), showing a very similar concept. Crystal formation has been already detailed in this work: <https://doi.org/10.1002/adfm.202001294>.

As a result, this work, despite of the interest for the community, deals with 1. a specific method; 2. a specific cation for 2D which is not part of the commonly used cations and 3. a method which has been already proven.

We thank the reviewer for bringing these publications to our attention. In the revised work, we cite these papers (pg. 4), but we explain how these 3 citations listed in this comment examine direct formation of 3D and 2D perovskites, or both, from their precursors in solution ((*Adv. Funct. Mater.* 31 (4), 2007473, *Chem. Mater.* 2019, 31, 22, 9472–9479 and *Adv. Funct. Mater.* 2020, 30, 2001294, respectively). We more clearly articulate the fact that these previously demonstrated *in situ* GIWAXS experiments do not give any insight into 2D/3D interface formation. We better explain that our *in situ* GIWAXS experiments focus on the *interconversion of pre-existing 3D perovskite into RDPs* following solution treatment, and not their direct synthesis from solution.

The 2D/3D heterostructures formed by exposing high-quality 3D perovskites to solutions containing 2D ligand cations is fundamentally different than direct synthesis of RDPs from precursors in solution. RDP thin films synthesized from solution consist of multidimensional grains with n values usually in the range of 5 – 20, and using these films as active layers in RDP-based solar cells has not yet achieved PCEs above 20%. The 2D/3D strategy as reported here allows for the formation of a thin ‘capping’ layer of 2D perovskites – usually comprised of $n = 1$ and 2 – atop a highly quality 3D perovskite active layer. This approach has been used extensively to make devices with efficiencies now exceeding 24%,^{5–7} including a recently certified 25.2% device,⁸ and so is highly general.

This method thus is new in reporting the formation mechanism of these 2D/3D interfaces. On page 4 of the manuscript, we have included:

“*In situ* GIWAXS has been used previously to study the formation of 2D or 3D perovskites directly from precursors in solution, tracking the formation of intermediate states and precursor complexes that eventually transform into perovskite.^{1,9-11} Here we instead use this technique to probe the transformation of the already fully-formed 3D perovskite surface into RDPs.”

For these reason there is not justified novelty nor general interest for publication in Nat. Comm. I would rather suggest a minor, more specialized Journal. In addition, the authors should provide at least a study case also for commonly used cations such as PEA or BAI. Importantly, the authors should also provide device statistics and stability data using consensus aging tests.

We now include device statistics from multiple devices, and find that the trends are consistent across the batch tested. We have included this new data in Figure S6 and Table S1 of the Supporting Information (they are also shown on Page 2 of the response letter for convenience). We have updated the values shown in Figure 4 of the main text to reflect the average values instead. The standard deviations for the different figures of merit are small (<1% in most cases). Due to the number of different film conditions and corresponding devices, we felt that stability aging tests were outside the scope of this study.

We agree with the referee that it will be beneficial in future to use *in situ* GIWAXS studies on other, more commonly used cations. We were unable to perform these experiments due to COVID-19 restrictions – the *in situ* experiments at the CLS require equipment and expertise from our lab, and we were unable to travel to the synchrotron to perform the measurements.

However, we posit that the results for the commonly used PEA ligand would be extremely similar, owing to the strong structural similarity between VBA and PEA. We have performed additional calculations on interfaces with PEA and BTA ligands, which support our hypothesis it is the π - π stacking at the interface that makes the interface formation more thermodynamically favourable (Figure S5 on page 2 of this response letter).

Reviewer #3 (Remarks to the Author):

The authors report a study on reduced dimensional perovskites (RDPs) on top of 3D perovskite thin films by performing in-situ GIWAXS. In short, they find that different rate of RDP formation and pathways (either formation of an intermediate phase or higher n layers first before eventually forming n=1 layers) depending on the 3D perovskite thin film composition and solvent choices. They correlate higher solar cell performance with slower formation rate of RDPs.

Based on the results and discussion presented in the manuscript, I cannot recommend publication for the following reasons:

In light of the reviewer's comments, we sought to further strengthen our findings with a new suite of morphological and photophysical characterizations. We now report AFM, SEM, ToF-SIMS, and PL quenching experiments that add new insights into how the different ligand solution solvents affect the surface morphology, and support our findings that slower 2D/3D transformations improve device performance by leading to thinner RDP layers that form conformally on the 3D perovskite grains, as detailed below.

1. It is not clear the solar cell results are statistically significant as there is no information on the number of devices and the distribution of relevant metrics.

We now include device statistics from multiple devices, and find that the trends are consistent across the batch tested. We have included this new data in Figure S6 and Table S1 of the Supporting Information (they are also shown on Page 2 of the response letter for convenience). We have updated the values shown in Figure 4 of the main text to reflect the average values instead. The standard deviations for the different figures of merit are small (<1% in most cases).

2. With the complex formation pathways and multiple factors changing together with different ligand solution combinations, it is unclear whether the relationship between slower formation rate of RDPs and higher solar cell performance can be established. For example, other than GIWAXS, there is no other structural characterization of the thin film. We do not know whether the treatment changes the grain size and etc. Moreover, other than solar cell results (which are convolution of many factors), there is no other electrical and optical characterization that support their claims.

Our initial work combined *in situ* GIWAXS, photovoltaic devices, and DFT to hypothesize that slower 2D/3D interface formation correlated with improved device performance, but as noted by the reviewer, lacked additional characterization to support this assertion.

To build on our findings from our previous submission, we have carried out several additional experiments to examine how the different rates of 2D/3D formation affect film morphology, RDP layer thickness, and carrier extraction. Below is an explanation of these experiments, the new insights they have given us into the influence of 2D/3D transformation on the film surface, and their significance in supporting the hypotheses of our previous submission. The below text is also included in pages 13 - 15 of the manuscript:

“We collected Atomic Force Microscopy (AFM) and Scanning Electron Microscopy (SEM) images to examine the influence of the different solvent conditions and corresponding 2D/3D transformations on the film surface morphology (Fig. 6 a, b). For films exposed to ligands dissolved in pure IPA, we observe large (micron-sized) unoriented flakes on the surface that are not conformal with the underlying 3D grains. Decreasing the concentration of the IPA reduces the amount of large unoriented crystallites and the morphology begins to resemble that of the 3D control film. For the lowest concentration of IPA using the 3:97 IPA:CF solution, the surface is highly similar to the 3D control (with slightly lower roughness), indicating that the RDPs are formed conformally on the surface. The 1:3 IPA:CB sample has the lowest surface roughness of the solvent series using 5 mg/mL VBABr solutions, and was also the only film in this series where we could observe an $n = 2 \rightarrow 1$ transition; whereas the 1 mM VBABr has an even lower surface roughness and was the only film to exhibit the $n = 3 \rightarrow 2 \rightarrow 1$ transitions. These observations are consistent with our hypothesis that lowering the concentration of IPA can slow down the growth of RDPs during 2D/3D interface formation: high concentrations of IPA causes rapid growth of large RDP grains that are unoriented relative to the 3D surface, whereas high concentrations of antisolvent instead facilitate conformal growth of the RDPs on the 3D grains. The slowest reaction rates may therefore yield higher quality RDPs through improved passivation and better surface conformity, which would explain the improved device performance.

Figure 6. (a) AFM height images and (b) SEM images of films treated with different ligand solutions. (c) PL lifetimes and fits for the same films with (quenched) and without (unquenched) a Spiro-OMeTAD layer spin-coated on top of the 2D/3D interfaces. The films were photoexcited (532 nm), and emission collected, on the substrate side of the film (glass / 3D perovskite interface). AFM and SEM images at different ranges / magnifications are shown in Fig. S6 and S7, and fitted PL lifetime parameters are shown in Table S2.

We performed Time-of-Flight Secondary Ion Mass Spectrometry (ToF-SIMS) to estimate the thickness of the RDP layers using depth profiles of the ligand-derived fragment $C_9H_{12}N^+$ (Fig. S8). We roughly estimate RDP layer thicknesses of 55 nm, 40 nm, 30 nm, and 35 nm for IPA:CB ratios

of 1:0, 3:1, 1:1, and 1:3, respectively and 4 nm for the 1 mM solution. This is indicative of the faster RDP growth rates forming thicker layers, which is known to hinder charge extraction from 2D/3D interfaces.¹²

To understand how the different 2D/3D interfaces affect charge extraction in photovoltaic devices, we performed photoluminescence (PL) lifetime quenching experiments by comparing lifetimes with and without a Spiro-OMeTAD quencher layer atop the 2D/3D interface (Fig. 6c).¹³⁻¹⁵ Aside from the 3D film, biexponential fits were required to fit the PL lifetimes. The values shown correspond to the longer of the two fitted lifetimes, which is most relevant for evaluating diffusion to and extraction at the quencher layer. The 3D film shows the strongest quenching, and the 1 mM VBABr film is also significantly quenched (and has the longest unquenched lifetime). For the remaining films, we observe that 2D/3D interfaces formed with higher concentrations of IPA have lifetimes that are even longer in the presence of the quencher, indicative of poor extraction of holes by Spiro-OMeTAD. This is consistent with our results from photovoltaic devices, where higher IPA concentrations lead to lower fill factors.”

Based on these reasons, I'm not convinced by authors' claims and cannot say I obtained much meaningful new insights. Therefore, I do not recommend publication.

We believe that the new suite of studies now included address the reviewer's feedback.

References

1. Quintero-Bermudez, R. *et al.* Compositional and orientational control in metal halide perovskites of reduced dimensionality. *Nat. Mater.* **17**, 900 (2018).
2. Fridriksson, M. B., van der Meer, N., de Haas, J. & Grozema, F. C. Tuning the Structural Rigidity of Two-Dimensional Ruddlesden–Popper Perovskites through the Organic Cation. *J. Phys. Chem. C* **124**, 28201–28209 (2020).
3. Smecca, E. *et al.* Stability of solution-processed MAPbI₃ and FAPbI₃ layers. *Phys. Chem. Chem. Phys.* **18**, 13413–13422 (2016).
4. Knight, A. J. *et al.* Halide Segregation in Mixed-Halide Perovskites: Influence of A-Site Cations. *ACS Energy Lett.* **6**, 799–808 (2021).
5. Jung, E. H. *et al.* Efficient, stable and scalable perovskite solar cells using poly(3-hexylthiophene). *Nature* **567**, 511 (2019).
6. Jiang, Q. *et al.* Surface passivation of perovskite film for efficient solar cells. *Nat. Photonics* **13**, 460–466 (2019).
7. Jeong, M. *et al.* Stable perovskite solar cells with efficiency exceeding 24.8% and 0.3-V voltage loss. *Science* **369**, 1615–1620 (2020).
8. Yoo, J. J. *et al.* Efficient perovskite solar cells via improved carrier management. *Nature* **590**, 587–593 (2021).
9. Szostak, R. *et al.* Revealing the Perovskite Film Formation Using the Gas Quenching Method by In Situ GIWAXS: Morphology, Properties, and Device Performance. *Adv. Funct. Mater.* **31**, 2007473 (2021).
10. Moral, R. F. *et al.* Synthesis of Polycrystalline Ruddlesden–Popper Organic Lead Halides and Their Growth Dynamics. *Chem. Mater.* **31**, 9472–9479 (2019).

11. Dong, J. *et al.* Mechanism of Crystal Formation in Ruddlesden–Popper Sn-Based Perovskites. *Adv. Funct. Mater.* **30**, 2001294 (2020).
12. Proppe, A. H. *et al.* Photochemically Cross-Linked Quantum Well Ligands for 2D/3D Perovskite Photovoltaics with Improved Photovoltage and Stability. *J. Am. Chem. Soc.* (2019) doi:10.1021/jacs.9b05083.
13. Lee, E. M. Y. & Tisdale, W. A. Determination of Exciton Diffusion Length by Transient Photoluminescence Quenching and Its Application to Quantum Dot Films. *J. Phys. Chem. C* **119**, 9005–9015 (2015).
14. Stranks, S. D. *et al.* Electron-Hole Diffusion Lengths Exceeding 1 Micrometer in an Organometal Trihalide Perovskite Absorber. *Science* **342**, 341–344 (2013).
15. Proppe, A. H. *et al.* Synthetic Control over Quantum Well Width Distribution and Carrier Migration in Low-Dimensional Perovskite Photovoltaics. *J. Am. Chem. Soc.* **140**, 2890–2896 (2018).

REVIEWERS' COMMENTS

Reviewer #1 (Remarks to the Author):

The authors have very significantly improved the manuscript and the main concern that I (and both other reviewers) had with the original version, the absence of statistical data for device characteristics, has been resolved. The same is true for most other points in my original review. While one may argue whether the results presented offer sufficient new insights to warrant publication in Nature Communications, I feel that the science presented is solid enough to be published.

Reviewer #2 (Remarks to the Author):

The authors have only partially replied to the Referee's comments. In particular, there is still no sufficient degree of novelty which does not justify publication in Nature Comm. The Giwax have been already reported as a routine method to study the formation of the 2D/3D systems (see for instance the works of the group of M.A. Loi) which contrasts to the authors statement mentioning that "This method thus is new in reporting the formation mechanism of these 2D/3D interfaces.". In addition, the concept of general validity and broad interest is weak given the absence of additional data on more standard cations. For these reasons I regret to say that this manuscript would be more appropriate for a more specialised Journal.

Reviewer #3 (Remarks to the Author):

The authors have addressed my comments and questions with significant amount of additional results in the revised manuscript. I recommend publication.

REVIEWERS' COMMENTS

Reviewer #1 (Remarks to the Author):

The authors have very significantly improved the manuscript and the main concern that I (and both other reviewers) had with the original version, the absence of statistical data for device characteristics, has been resolved. The same is true for most other points in my original review. While one may argue whether the results presented offer sufficient new insights to warrant publication in Nature Communications, I feel that the science presented is solid enough to be published.

We thank the reviewer for their reconsideration of our work, and their recommendation for it to be published.

Reviewer #2 (Remarks to the Author):

The authors have only partially replied to the Referee's comments. In particular, there is still no sufficient degree of novelty which does not justify publication in Nature Comm. The Giwax have been already reported as a routine method to study the formation of the 2D/3D systems (see for instance the works of the group of M.A. Loi) which contrasts to the authors statement mentioning that "This method thus is new in reporting the formation mechanism of these 2D/3D interfaces.". In addition, the concept of general validity and broad interest is weak given the absence of additional data on more standard cations. For these reasons I regret to say that this manuscript would be more appropriate for a more specialised Journal.

As stated in our previous response letter, we were unable to provide additional data on more standard cations due to COVID-19 lockdown restrictions preventing us from accessing the synchrotron. From DFT calculations, we show that this interface stabilization occurs for BTA, PEA, and VBA molecules, indicative of the generality of our findings. Moreover, the structure of VBA is highly similar to PEA, a more commonly used cation, so it is reasonable to expect the results reported here to be similar for PEA.

We again disagree that GIWAXS has been used to studying 2D/3D interface formation from conversion of the 3D perovskite surface. This is fundamentally different than the direct synthesis of mixed 2D/3D thin films, which have been studied using *in situ* GIWAXS (including by our group). To the best of our knowledge, there is no such report for 3D-to-2D interconversion, and twice now the referee has not provided an example.

Reviewer #3 (Remarks to the Author):

The authors have addressed my comments and questions with significant amount of additional results in the revised manuscript. I recommend publication.

We thank the reviewer for their reconsideration of our work, and their recommendation for it to be published.